# Substance use among young people in the West Arsi Zone, Ethiopia: A cross-sectional study

**Jemal Ebrahim Shifa**[1,2*], **Jon Adams**[1], **Daniel Demant**[1,3]

**1** Faculty of Health, School of Public Health, University of Technology Sydney, New South Wales, Sydney Australia, **2** Department of Psychiatry, Faculty of Health Sciences, Madda Walabu University, Shashemene, Ethiopia, **3** Faculty of Health, School of Public Health and Social Work, Queensland University of Technology, Queensland, Brisbane, Australia

* jemalebrahim.shifa@student.uts.edu.au, jemebrie@gmail.com

## Abstract

### Introduction

Substance use is a pressing public health concern in young Ethiopians, impacting their physical, psychosocial, and emotional well-being and productivity. However, there is a limited understanding of the prevalence and factors associated with substance use in this population both across Ethiopia and in the West Arsi zone specifically. This study investigates the prevalence of substance use and associated factors among young people in the West Arsi Zone, Ethiopia.

### Methods

A community-based cross-sectional survey was conducted among 427 randomly selected young people aged 14–29 in the West Arsi zone of the Oromia region, Ethiopia. Data were collected using structured interviewer-administered questionnaires. Logistic regression analysis was performed to determine the association between the outcome and independent variables. Ethical approval was obtained from the University of Technology Sydney, Australia, and Madda Walabu University, Ethiopia.

### Results

A total of 424 participants were included in the analysis, giving a response rate of 99.3%. The overall lifetime prevalence of any substance use among the study participants was 48.1% (95% CI: 43.3%, 53.0%) and the prevalence of current substance use was 72.5% (95% CI: 65.9, 78.5). Among lifetime users, 76.5% reported chewing khat, 49.0% drinking alcohol, 33.3% using various forms of tobacco, and 23.0% using cannabis. Being male, having a single marital status, a family history of substance use, low perceived social support, and the presence of mental health conditions were associated with an increased likelihood of substance use.

**Data availability statement:** All relevant data used in this study are fully included within the

paper and its Supporting Information files (S1 Table).

**Funding:** JE has received a Higher Degree by Research (HDR) support funds from the University of Technology Sydney (UTS) for study data collection process. UTS had no role in the study design, analysis, and interpretation of the data, in the writing of the manuscript, or in the decision to submit the article for publication.

**Competing interests:** The authors have declared that no competing interests exist.

## Conclusions

About half of the study participants reported a history of use of at least one substance from alcohol, khat, tobacco, or cannabis in their lifetime, highlighting the need for appropriate focused interventions to help address the growing challenges of substance use amongst young people in Ethiopia.

## Introduction

The use of psychoactive substances such as alcohol, khat, tobacco, and cannabis remains a public health concern. Globally, an estimated 2.3 billion people consume alcohol [1], 1.3 billion tobacco products [2], 147 million cannabis [3], and 20 million khat [4]. About half a million global deaths are attributable to substance use annually, and substance use contributes 1.3 percent to the global burden of disease [5].

Substance use is associated with various health risks which, when left untreated, have the potential to lead to a range of personal and social problems [6,7]. These, in turn, have a wider societal impact through lost productivity, early death, increased healthcare spending as well as increased expenses for criminal justice and social welfare [6].

Substance use and its consequences are particularly pronounced in young people [8–11]. Within this demographic, substance use manifests as a complex phenomenon driven by multifaceted factors. Young people use substances to enhance positive emotions, foster connections or fit into social groupings, conform to social norms or peer expectations, and as a maladaptive coping mechanism for social, emotional, or physical health issues [12]. Furthermore, young individuals may use substances to experiment, to assert their independence, or to feel more mature [12,13].

While substance use is a global concern, some countries are disproportionally affected. In Ethiopia, a country marked by economic hardships, political tensions, and stark social disparities, young people may resort to substances like alcohol, khat, tobacco, or cannabis to overcome these stressors. In Ethiopian culture, chewing khat and drinking alcohol is commonly seen as a way to connect with others during social gatherings and social ceremonies, and to foster a sense of belonging with the local community [14,15]. Furthermore, Ethiopia has a long history of cultivating, producing, trading, and consuming various homegrown plants and herbs with psychoactive properties such as catha edulis (known as khat/chat in Ethiopia), cannabis (known as hashish or ganja in Ethiopia), and alcohol (locally produced tella, tej, and areke) [7,16–18].

The growing urbanization and globalization in Ethiopia and increasing exposure to global consumer culture contribute to shifting societal norms, potentially increasing risks associated with increased substance use [19–21]. Further, perceptions that khat and tobacco boost concentration and working memory, that khat breaks alcohol intoxication, and that cigarettes complement the alcohol effect, have all contributed to increased substance use among young people in Ethiopia [15,22,23].

Despite the widespread nature of substance use, previous studies on this matter in Ethiopia have predominantly concentrated on schools or higher education settings, providing insights into the prevalence and determinants of substance use among students. For instance, four reviews conducted in recent years have estimated the magnitude of substance use and its determinants among young people attending schools [18,19,24,25], while only one addresses the problem among young people, aged 10–24 years in the general community [26]. There are also limitations due to a lack of utilizing standardized tools to assess substance use, and a lack of coverage for the wider young population, especially in rural or semi-urban areas.

Thus, there is a need to comprehensively examine the prevalence of substance use among young people beyond educational settings, encompassing a diverse range of substances and socio-cultural contexts, covering both adolescents and youth in the general community. In direct response to this gap, this cross-sectional study aims to examine the prevalence of substance use and associated factors among young people in the west Arsi zone, Ethiopia. Such work is important as one means by which to help inform, design, and implement targeted interventions and public health policies on this topic in Ethiopia [18, 19].

## Materials and methods

### Study design

A cross-sectional study was conducted from May 18, 2023, to September 22, 2023, to assess the prevalence of substance use and associated factors among young people in the West Arsi zone.

### Study setting

The study was conducted in the West Arsi zone in the Oromia Regional State. The zone comprises 13 districts (woredas) and two town administrations, Ethiopia, which comprises 13 districts (woredas) and three town administrations. In 2022, the total population of this zone was approximately three million, with young people (aged 10–29 years) accounting for 1.2 million [27].

The region is known for its substantial production and distribution of various psychoactive substances, including alcohol, khat, cannabis, and tobacco, which are both widely available and culturally integrated. Additionally, West Arsi experiences a high unemployment rate, reflecting the broader economic challenges across Ethiopia [28], which may further heighten vulnerability. Healthcare infrastructure in West Arsi, particularly for mental health or substance-related services, is limited, restricting access to necessary support. Given these combined factors, assessing substance use among young people in this setting is important to inform targeted interventions and policy responses.

### Study population and recruitment

The study considered all young people living in the West Arsi Zone of the Oromia region, Ethiopia, as a source population. The target population included young people aged 14–29 years, residents of the study area who had lived there for at least six months at the time of the survey (In the Ethiopian context, six months is widely regarded as the minimum period required for recognizing residency in a neighborhood), could provide informed consent and/or assent, and were willing to participate. Young people who were critically ill and unable to provide information and individuals absent during the data collection period were excluded.

The sample size for this study was determined using a 95% CI with a Z-value of 1.96, a 5% level of precision (d = 0.05), an estimated proportion of substance use (P = 50%), and a 10% non-response rate. The proportion of 50% was considered to ensure the inclusion of the maximum possible sample size for this study. Based on these parameters, the required sample size was determined to be 427 participants.

To further ensure adequate statistical power and account for key factors associated with substance use, such as sex, family history of substance use, and peer substance use reported in previous literature, additional sample size calculations were conducted using the double population proportion formula. However, a comparison of all independently calculated sample sizes revealed that the single proportion formula based on a 50% prevalence provided the largest sample size [29]. Consequently, the sample size of 427 participants was adopted as the final sample size for this study.

The study was conducted in the West Arsi zone of the Oromia Regional State, Ethiopia. A convenient and purposive sampling method was applied to select the study setting. This zone was chosen due to the researcher's familiarity with the area, the significant production, distribution, and accessibility of various psychoactive substances, and the high unemployment rate in the region [28], which has been identified as a potential risk factor for substance use.

Among the 13 woredas in the West Arsi zone, four woredas were randomly selected, representing approximately 30% of the woredas. This proportion is considered scientifically sound for ensuring representativeness while maintaining feasibility. From each selected woreda, four kebeles (the smallest administrative unit in Ethiopia) were also selected based on their accessibility and the concentration of the young population, with two kebeles drawn from urban and two from rural areas (S1 Table). The urban-rural stratification was included to facilitate comparisons, given documented variations in substance use between urban and rural communities in Ethiopia [30].

Following the initial step that involved the identification of the selected kebeles, and to prepare for data collection, a one-week census of households with young individuals was conducted. Each household was given a unique number to establish a sampling frame. The numbering process began at a central or predefined landmark in each kebele, such as the kebele administration office, mosque, church, school, or another prominent feature depending on the setting. From this starting point, households were numbered systematically, ensuring full coverage within the kebele boundary until the last household to be included in the study was reached.

Local young individuals, capable of writing, participated voluntarily in the process of household identification and numbering, assisted by a local law enforcement personnel assigned from the kebele administration office to ensure the safety and smooth execution of the process, particularly to the local security concerns. This was necessary due to the challenge of obtaining comprehensive information about the target population from local government sources.

The unique household identification number formed the basis for a lottery system used to randomly select households for participation. Samples were proportionally allocated among the selected kebeles to ensure representation based on the total number of young people in each kebele. Once households were identified, data collectors conducted interviews with young individuals in the selected households. In situations where more than one eligible individual was present in a household, a lottery method was applied to select one participant, ensuring randomization and avoiding clustering effects.

## Data collection process

A face-to-face interview was conducted door-to-door to administer the questionnaire. The overall data collection took place from May 18, 2023, to September 22, 2023. Face-to-face data collection was adopted to counter the potential for information contamination due to the low literacy rate in the region [31]. Experience with research in the Ethiopian context shows that, in the case of remotely administered and self-reported questionnaires, young people tend to share information with their friends and relatives leading to potentially biased responses in addition to ethical concerns. This information sharing, a critical problem in research, could result from low literacy, the unfamiliarity of the community with the concept of research, and communication problems that interfere with information comprehension. Additionally, online surveys were not considered viable due to the limited and inconsistent internet access and use across the country. Given these challenges, face-to-face interviews were determined to be the most appropriate and effective method for data collection, ensuring reliable and accurate responses from the study participants.

The questionnaire used in the survey data collection was structured into sections, including items (I) assessing the sociodemographic characteristics of respondents, (II) screening substance use, III) assessing mental health conditions, and IV) examining perceived social support.

Most of the tools used in the study were standardized questionnaires with prior validation in Ethiopia and other African settings. Previously translated versions of these tools, used in various earlier studies, were adapted for the current study. For some tools, additional translation from Amharic to Afan Oromo was necessary to cater to the local language needs of the study site. This translation process was conducted collaboratively by the primary researcher, an expert from Madda Walabu University's English department, and the data collectors. The process included iterative back-and-forth translation to ensure linguistic consistency and cultural relevance. This meticulous approach ensured that the questionnaires captured the intended information while respecting the cultural and linguistic nuances of the target population.

Data collection was carried out by a team of four experts who were proficient in both Afan Oromoo and Amharic. They were assisted by local law enforcement personnel assigned by the Kebele administration office to address the safety concerns and smooth implementation, particularly in areas with potential security concerns. The primary researcher supervised the overall data collection process, providing oversight and assisting in data collection when necessary.

Before data collection began, a training session was conducted for the data collectors to ensure a shared understanding of the study objectives, the data collection tools, and the overall process.

During data collection, participants were given a detailed explanation of the study as included in the information sheet, and those meeting the eligibility criteria were asked for informed consent. Most participants provided written consent, while those with functional illiteracy gave verbal consent. For participants under the age of 18, written consent was obtained from parents or caregivers after explaining the study's purpose in detail, as outlined in the translated parent information sheet. Participants under 18 also provided written consent, apart from three individuals within this age group who declined to do so. The reason cited for declining consent was a privacy concern, stating that they did not wish to disclose their substance use to their families. Their decisions were respected, and the next eligible participants were included instead.

Interviews were conducted in locations chosen based on the preferences of participants, ensuring a comfortable and private setting. The duration of each interview ranged from 35 to 60 minutes.

## Variables and measurements

### Substance use

The primary outcome variable was substance use, including alcohol, khat, tobacco, and cannabis, measured in the lifetime and current use categories. Lifetime substance use referred to the use of substance at least once in a lifetime whereas current substance use was defined as the use of any substances within the past 30 days preceding the survey.

In this study, the variable *substance use* was derived from the specific types of substances consumed, including alcohol, khat, tobacco, and cannabis. This dichotomous variable, *substance use,* with yes/no options, was subsequently created, and participants were classified as substance users if responded "yes" to having used at least one of the listed substances, and as nonusers if they responded "no" to all listed substances.

Each substance was measured using a combination of locally validated standard tools alongside certain author-developed questions tailored to align with the cultural and contextual specifics of the study, detailed as follows;

*Alcohol*- to assess alcohol use, we initially asked two straightforward and general questions with "yes" or "no" response options, which were also adapted to screen other substances—khat, tobacco, and cannabis use. These questions were as follows:

- Lifetime use: Have you ever drunk [alcohol/chewed khat/used cannabis]?

- Current use (past 30 days): Are you currently drinking [alcohol/chewing khat/using cannabis]?

The Alcohol Use Disorders Identification Test (AUDIT) was used to determine the risk profile of participants who reported consuming alcohol in the initial assessment. The AUDIT was developed by the World Health Organization (WHO) for screening excessive alcohol consumption and to assist in brief assessments [32], and has been validated for use in Ethiopia, as well as across various genders, target populations, and a diverse range of racial and ethnic groups ([32–35].

The AUDIT comprises 10 items to evaluate alcohol consumption, drinking patterns, and alcohol-related issues within the preceding 12 months. Each item is rated on a four-point scale, and the cumulative score ranges from zero to 40. Scores between 1 and 7 indicate low risk, scores from 8 to 14 suggest hazardous alcohol use and a score of 15 or higher indicates alcohol dependence. Typically, a score of eight or above serves as the threshold for diagnosing alcohol use disorder [36]. The current study obtained a good level of internal consistency, Cronbach's α = 0.87, in line with a previous local study [37].

*Khat*- among those who indicated using khat in the "yes/no" screening questions for lifetime and current khat use, the Problematic Khat Use Screening Test (PKUST-17) was administered to examine problematic use. The PKUST-17 was recently developed and validated in the Ethiopian context [23], and contains 17 items, with response options on a 5-point-Likert-scale, resulting in an overall score ranging from 0–68 [23]. Findings were presented based on the overall median score. Individuals scoring below the median were categorized as having lower problematic khat use, while those scoring equal to or above the median were considered to exhibit higher problematic khat use. The Cronbach's alpha in our study was excellent (0.96) and similar to the 0.93 reported in a previous Ethiopian study [23].

*Tobacco*- to assess tobacco use, we adopted the definition of "tobacco use" as defined by the Ethiopian Demographic and Health Survey (EDHS), that an individual is classified as a "tobacco user" if they report using any of the various forms or types of tobacco listed in the EDHS database such as cigarette smoking, piped tobacco smoking, chewing tobacco, snuff/suret, shisha, gaya (local traditional smoking tobacco leaves) or any other type [38, 39]. The use of any form of tobacco was assessed by asking respondents about their ever and current use of any form of tobacco.

*Cannabis*- to measure the lifetime and current use of cannabis among study participants, a simple yes/no question developed by the authors was utilized. Participants were classified as "cannabis users" if they responded 'yes' to the cannabis screening question and as "nonusers" if they responded 'no'.

## Mental health conditions

Mental health conditions were assessed using previously used and validated instruments: the Patient Health Questionnaire (PHQ-9) for depression [40,41], the Generalized Anxiety

Disorder (GAD-7) for anxiety disorder [42], the (Trauma Screening Questionnaire) TSQ-10 for Post-traumatic Stress Disorder (PTSD) [43], and items from previous studies on suicide behavior [44]. These tools are standard and have demonstrated reliability across various demographics and settings [34,45,46]. In this study, participants were classified as positive for mental health conditions if they responded affirmatively to at least one of these conditions, and negative otherwise.

## Perceived social support

Social support was measured using the 12-item Multidimensional Scale of Perceived Social Support (MSPSS), previously used in Ethiopia and other similar settings across Africa. The MSPSS has demonstrated strong internal consistency (Cronbach's alpha values typically above 0.8) and test-retest reliability, making it a robust instrument for assessing perceived social support across different cultural and demographic contexts [47]. This study followed adaptations from comparable settings in Africa, such as Uganda and Malawi, where slight language and context adjustments were made to preserve the instrument's cultural relevance while maintaining its original three-factor structure [48, 49]. In this study, perceived social support levels were classified as low, moderate, and high based on the mean score [50].

## Other demographic, social, and family factors

Other independent variables such as age, sex, education, religion, ethnicity, marital status, occupation, place of residence, children status, number of children, family size, spouse education, spouse education, family income, and source of income, family history of substance use, and family history of mental illness were assessed using a combination of author-constructed questions and adapted items from previous studies [51, 52].

## Data management and statistical analysis

Data analysis was carried out by using Stata, Version 18.0. Variables were described using frequency, percentage, and tables. For continuous variables, normality was checked using histograms, and median and inter-quarter ranges were reported.

Simple binary logistic regression analysis for each independent variable was performed against the dependent variable to see the impact of each factor on substance use among young people, without adjusting for the effect of other variables.

Independent variables found to be significant in the simple binary logistic regression analysis at a cut-off point of p-value < 0.25 with a 95% confidence interval were included in a multiple binary logistic regression model. In the multiple binary logistic regression model, the effect of each independent variable on substance use was assessed by controlling for the possible confounders. Factors that remained insignificant in the final model at a p-value above 0.05 were removed.

The occurrence of multicollinearity was checked for the final model with an overall cut-off point of variance inflation factor (VIF) less than five. The overall fitness of the model was assessed using a chi-squared test. The associated p-value indicates the model is statistically significant (Prob>chi2 = 0.0000), suggesting that the predictors collectively affect the dependent variable- any substance use. The Receiver Operating Characteristic (ROC) curve analysis was employed to evaluate the predictive performance of the model utilized and the value of the area under the curve (AUC) falls within the ranges suggesting good discriminatory ability supporting the utility of our model in predicting the substance use among young people in Ethiopia.

## Ethical considerations

The study was conducted in accordance with the Declaration of Helsinki's ethical principles and guidelines, ensuring that all potential participants were fully informed about the study and given enough time to make informed decisions about participation. Ethics approval was obtained from the Health and Medical Research Ethics Committee at the University of Technology Sydney (reference number: ETH23-7890) and the Campus Research Ethics Review Committee of Madda Walabu University, Shashemene Campus, Ethiopia (reference number: RCSTT/34/2015). A support letter was obtained from Madda Walabu University and Woreda administration offices.

## Result

### Study sample characteristics

Table 1 shows the sociodemographic characteristics of the N = 424 participants included in the analysis, with a response rate of 99.3%. The sample was comprised of 63.9% (n = 271) males and 36.1% (n = 153) females, with a median age of 23 years. About one-third (34.7%, n = 147) had attended basic formal education (grades 1–8), 58.0% (n = 246) were single, and 37.5% (n = 159) were employed. The majority identified as Oromo 71.9% (n = 305) in ethnicity, over half were Muslim, while 20.8% (n = 88) reported low perceived social support.

### Distribution of substance use by demographic characteristics

Overall, 48.1% (204/424) of participants reported having used substances in their lifetime, and among them, 72.5% (148/204) were currently using substances (within the 30 days preceding the survey). A gender difference in lifetime substance use was observed, with 79.1% (n = 163) of males reporting use compared to 20.9% (n = 41) of females. Additional analyses looking at the relationship between various demographic and social factors and the prevalence of substance use among participants are provided in (Table 3).

Disparities in the levels of substance use among specific demographics and social groups were also identified, including individuals aged 25–29 years old (48.5%), singles (64.2%), employed individuals (41.7%), those with basic formal education (grades 1–8) (28.9%), urban residents (56.9%), and those identifying as Muslim (43.6%), when compared to their counterparts. Additionally, participants who reported having children, those with moderate perceived social support status, a family history of any substance use, and those belonging to medium-sized families (4–6 members) had relatively higher levels of substance use than their respective counterparts. Among the study participants who reported substance use, 16.7% (n = 34) had a family history of mental illness, while 83.3% (n = 170) did not. Similarly, among the study participants who reported substance use, 47.1% (n = 96) reported the presence of any mental health condition, while 52.9% (n = 108) did not (Table 2).

### Substance use prevalence

In this study, the overall lifetime and current prevalence of any substance use among young people were 48.1% (95% CI: 43.3%, 53.0%) and 72.5% (95% CI: 65.9, 78.5), respectively.

The lifetime and current prevalence of alcohol among the study participants was 49.0% (95% CI: 42.0, 56.0) and 80.0% (95% CI: 70.8, 87.3), respectively. In a further analysis of the alcohol use risk levels in the last 12 months, assessed through the AUDIT-10 items among young individuals, 48.0% (n = 48), 30.0% (n = 30), and 22.0% (n = 22) exhibited low risk, probable hazardous use, and probable alcohol dependence, respectively (Table 3).

The lifetime and current prevalence of khat use among the study participants was 76.5% at (95% CI: 70.0, 82.0) and 91.7% (95% CI: 86.2, 95.5), respectively. The study identified

**Table 1. Distribution of the sociodemographic characteristics of the study sample, Ethiopia, 2023, (n = 424).**

| Variable | Frequency | Percentage |
|---|---|---|
| Sex: | | |
| Female | 153 | 36.1 |
| Male | 271 | 63.9 |
| Age in years: | | |
| 14–19 | 103 | 24.3 |
| 20–24 | 124 | 29.2 |
| 25–29 | 197 | 46.5 |
| Participant Education: | | |
| Not attended formal education | 61 | 14.4 |
| Attended basic formal education | 147 | 34.7 |
| Attended secondary school | 119 | 28.0 |
| Attended higher education | 97 | 22.9 |
| Marital Status: | | |
| Single | 246 | 58.0 |
| Ever married | 178 | 42.0 |
| Participant Occupation: | | |
| Farmer | 62 | 14.6 |
| Employed | 159 | 37.5 |
| Student | 104 | 24.5 |
| Unemployed | 99 | 23.4 |
| Place of residence: | | |
| Rural | 193 | 45.5 |
| Urban | 231 | 54.5 |
| Ethnicity: | | |
| Oromo | 305 | 71.9 |
| Amharic | 40 | 9.4 |
| Sidama | 46 | 10.9 |
| Wolaita | 18 | 4.3 |
| Others | 15 | 3.5 |
| Religion: | | |
| Orthodox | 87 | 20.5 |
| Muslim | 221 | 52.1 |
| Protestant | 105 | 24.8 |
| Catholic | 11 | 2.6 |
| Spouse Education: | | |
| Not attended formal education | 32 | 18.0 |
| Attended basic formal education | 56 | 31.5 |
| Attended secondary school | 54 | 30.3 |
| Attended higher education | 36 | 20.2 |
| Spouse Occupation: | | |
| Farmer | 35 | 19.7 |
| Employed | 87 | 48.9 |
| Student | 4 | 2.2 |
| Unemployed | 52 | 29.2 |
| Children Status: | | |
| No | 276 | 65.1 |
| Yes | 148 | 34.9 |

*(Continued)*

**Table 1.** (Continued)

| Variable | Frequency | Percentage |
|---|---|---|
| Number of children: | | |
| Less than 4 children | 151 | 84.8 |
| Greater than 4 children | 27 | 15.2 |
| Household size: | | |
| Small family | 98 | 23.1 |
| Medium family | 208 | 49.1 |
| Large family | 118 | 27.8 |
| Family income per year: | | |
| Low income | 161 | 38.0 |
| Medium income | 179 | 42.2 |
| High income | 84 | 19.8 |
| Source of family income: | | |
| Agriculture | 191 | 45.1 |
| Trade | 57 | 13.4 |
| Private business | 83 | 19.6 |
| Salary | 93 | 21.9 |
| Level of social support: | | |
| Low perceived social support | 88 | 20.8 |
| Moderate perceived social support | 299 | 70.5 |
| High perceived social support | 37 | 8.7 |
| Family history of substance use: | | |
| No | 292 | 68.9 |
| Yes | 132 | 31.1 |
| Family history of mental illness: | | |
| No | 374 | 88.2 |
| Yes | 50 | 11.8 |
| Any mental health condition: | | |
| No | 271 | 63.9 |
| Yes | 153 | 36.1 |

an overall mean score of 1.28 of the PKUST-17 for problematic khat use among the study participants. Using this mean score as the reference for categorization into higher and lower problematic use, the findings indicated that 56.4% (n = 88) reported low problematic khat use, while 43.6% (n = 68) reported high problematic khat use (Table 3).

The lifetime prevalence of tobacco use was 33.3% (95% CI: 26.9, 40.2), while the current prevalence was 76.5% (95% CI: 64.6, 85.9) (Table 3). Lifetime tobacco use was calculated among individuals who reported lifetime use of any substance in the list (68/204), whereas current use was derived from lifetime users (52/68). Notably, both the current and lifetime prevalence of substance use were higher among males, individuals in the age group 20–24 years old, and those who reported at least one mental health condition, as detailed in (Table 3). Among the lifetime tobacco users, 47% smoked cigarettes, 44% smoked shisha, and the remaining used other forms of tobacco. Among current tobacco users, 51.9% (n = 27) reported smoking cigarettes, 38.5% (n = 20) used shisha, 1.9% (n = 1) used in the form of snuff (chewing tobacco), 3.8% (n = 2) used both cigarettes and shisha and 3.8% (n = 2) reported using both cigarettes and chewing tobacco.

**Table 2. The relationship between various sociodemographic factors and substance use among the study sample, Ethiopia, 2023, (n = 424).**

| Variable | Level | Total | No | Yes | P-value |
|---|---|---|---|---|---|
| N | | 424 | 220 (51.9%) | 204 (48.1%) | |
| Sex of participants | Female | 153 | 112 (50.9%) | 41 (20.1%) | <0.001 |
| | Male | 271 | 108 (49.1%) | 163 (79.9%) | |
| Age of the participant | 14-19 | 103 | 64 (29.1%) | 39 (19.1%) | 0.050 |
| | 20–24 | 124 | 58 (26.4%) | 66 (32.4%) | |
| | 25–29 | 197 | 98 (44.5%) | 99 (48.5%) | |
| Participant marital status | Single | 246 | 115 (52.3%) | 131 (64.2%) | 0.013 |
| | Ever married | 178 | 105 (47.7%) | 73 (35.8%) | |
| Place of residence | Rural | 193 | 105 (47.7%) | 88 (43.1%) | 0.343 |
| | Urban | 231 | 115 (52.3%) | 116 (56.9%) | |
| Participant occupational status | Farmer | 62 | 35 (15.9%) | 27 (13.2%) | 0.168 |
| | Employed | 59 | 74 (33.6%) | 85 (41.7%) | |
| | Student | 104 | 62 (28.2%) | 42 (20.6%) | |
| | Unemployed | 99 | 49 (22.3%) | 50 (24.5%) | |
| Spouse occupational status | Farmer | 35 | 25 (23.8%) | 10 (13.7%) | 0.327 |
| | Employed | 87 | 51 (48.6%) | 36 (49.3%) | |
| | Student | 4 | 2 (1.9%) | 2 (2.7%) | |
| | Unemployed | 52 | 27 (25.7%) | 25 (34.3%) | |
| Participant Educational Status | Not attended formal education | 61 | 26 (11.8%) | 35 (17.2%) | 0.020 |
| | Attended basic formal education | 147 | 88 (40.0%) | 59 (28.9%) | |
| | Attended secondary school | 119 | 65 (29.5%) | 54 (26.5%) | |
| | Attended higher education | 97 | 41 (18.6%) | 56 (27.4%) | |
| Spouse Educational Status | Not attended formal education | 32 | 17 (16.2%) | 15 (20.5%) | 0.290 |
| | Attended basic formal education | 56 | 29 (27.6%) | 27 (37.0%) | |
| | Attended secondary school | 54 | 37 (35.2%) | 17 (23.3%) | |
| | Attended higher education | 36 | 22 (21.0%) | 14 (19.2%) | |
| Children Status | No | 276 | 132 (60.0%) | 144 (70.6%) | 0.022 |
| | Yes | 148 | 88 (40.0%) | 60 (29.4%) | |
| Number of children | < 4 children | 151 | 87 (82.9%) | 64 (87.7%) | 0.379 |
| | ≥ 4 children | 27 | 18 (17.1%) | 9 (12.3%) | |
| Household size | Small family | 98 | 43 (19.5%) | 55 (27.0%) | 0.174 |
| | Medium family | 208 | 115 (52.3%) | 93 (45.6%) | |
| | Large family | 118 | 62 (28.2%) | 56 (27.4%) | |
| Family income per year | Small income | 161 | 78 (35.4%) | 83 (40.2%) | 0.538 |
| | Medium income | 179 | 97 (44.1%) | 82 (40.7%) | |
| | High income | 84 | 45 (20.5%) | 39 (19.1%) | |
| Source of family income | Agriculture | 191 | 103 (46.8%) | 88 (47.7%) | 0.069 |
| | Trade | 57 | 25 (11.4%) | 32 (15.7%) | |
| | Private business | 83 | 51 (23.2%) | 32 (15.7%) | |
| | Salary | 93 | 41 (18.6%) | 52 (25.5%) | |
| Participant religion | Orthodox | 87 | 28 (12.7%) | 59 (28.9%) | <0.001 |
| | Muslim | 221 | 132 (60.0%) | 89 (43.6%) | |
| | Protestant | 105 | 57 (25.9%) | 48 (23.5%) | |
| | Catholic | 11 | 3 (1.4%) | 8 (3.9%) | |

*(Continued)*

**Table 2.** (Continued)

| Variable | Level | Total | No | Yes | P-value |
|---|---|---|---|---|---|
| Ethnicity | Oromo | 305 | 167 (75.9%) | 138 (67.6%) | 0.388 |
| | Amhara | 40 | 16 (7.3%) | 24 (11.8%) | |
| | Sidama | 46 | 22 (10.0%) | 24 (11.8%) | |
| | Wolaita | 18 | 8 (3.6%) | 10 (4.9%) | |
| | Others | 15 | 7 (3.2%) | 8 (3.9%) | |
| Level of social support | Low social support | 88 | 18 (8.2%) | 70 (34.3%) | <0.001 |
| | Moderate social support | 299 | 172 (78.2%) | 127 (62.3%) | |
| | High social support | 37 | 30 (13.6%) | 7 (3.4%) | |
| Any mental health condition | No | 271 | 163 (74.1%) | 108 (52.9%) | <0.001 |
| | Yes | 153 | 57 (25.9%) | 96 (47.1%) | |
| Family history of substance use | No | 292 | 201 (91.4%) | 91 (44.6%) | <0.001 |
| | Yes | 132 | 19 (8.6%) | 113 (55.4%) | |
| Family history of mental illness | No | 374 | 204 (92.7%) | 170 (83.3%) | 0.001 |
| | Yes | 50 | 16 (7.3%) | 34 (16.7%) | |

The lifetime and current prevalence of cannabis use was 23.0% (95% CI: 17.4, 29.4) and 68.1% (95% CI: 52.9, 80.9) respectively. Among the current users in the study, 87.5% (n = 28) were males, 40.6% (n = 13) were in the age range of 25–29 years old, and 34.4% (n = 11) had completed basic formal education (grades 1–8). Additionally, over half of the current cannabis user samples reported low perceived social support, while approximately two-thirds reported at least one mental health condition (Table 3).

## Factors associated with substance use

In the preliminary unadjusted bivariate model analysis, various covariates were examined independently for their potential association with substance use. These covariates included sex, age, marital status, education, occupation, religion, place of residence, level of social support, mental health condition, family size, source of income, family history of any substance use, and family history of mental illness (Table 4).

In the adjusted model analysis controlling for potential confounders, participant sex, marital status, family history of substance use, perceived social support, and the presence of any reported mental health condition remained significantly associated with the likelihood of substance use (Table 4).

Notably, male study participants exhibited a significantly higher likelihood (6.37 times higher odds) of substance use compared to their female counterparts, showing gender-based disparities in substance use (AOR = 6.37, 95%CI:3.35, 12.12, p < 0.001).

Marital status also played a significant role with singles demonstrating a significantly higher likelihood (2.12 times) of substance use in comparison to those who were ever married (AOR = 2.12, 95%CI: 1.06, 4.23, p = 0.033).

Individuals with low perceived social support had approximately 6.5 times higher odds of any substance use compared to those with moderate and high perceived social support (AOR = 6.55, 95%CI:2.02, 21.30, p < 0.001).

Participants who reported a history of substance use in their family had 11.66 times higher odds of substance use than those with no family history of substance use (AOR = 11.66, 95%CI: 6.04, 22.55, p < 0.001). Furthermore, young individuals with any reported mental

**Table 3. Lifetime and current substance use distribution by various sociodemographic characteristics of the study samples, Ethiopia 2023, (n = 424).**

| Variable | Current use (YES) | | | | Ever use (YES) | | | |
|---|---|---|---|---|---|---|---|---|
| | Alcohol | Khat | Tobacco | Cannabis | Alcohol | Khat | Tobacco | Cannabis |
| Sex: | | | | | | | | |
| Female | 18(22.5%) | 21(14.7%) | 4(7.7%) | 4(12.5%) | 27(27.0%) | 26(16.7%) | 7(10.3%) | 6(12.8%) |
| Male | 62(77.5%) | 122(85.3%) | 48(92.3%) | 28(87.5%) | 73(73.0%) | 130(83.3%) | 61(89.7%) | 41(87.2%) |
| Age in years: | | | | | | | | |
| 14–19 | 14(17.5%) | 30(21.0%) | 6(11.5%) | 6(18.8%) | 15(15.0%) | 33(21.2%) | 9(13.3%) | 9(19.1%) |
| 20–24 | 25(31.3%) | 44(30.8%) | 14(26.9%) | 13(40.6%) | 33(33.0%) | 49(31.4%) | 23(33.8%) | 18(38.3%) |
| 25–29 | 41(51.2%) | 69(48.2%) | 32(61.6%) | 13(40.6%) | 52(52.0%) | 74(47.4%) | 36(52.9%) | 20(42.6%) |
| Participant Education | | | | | | | | |
| Not attended formal education | 13(16.2%) | 22(15.4%) | 15(28.8%) | 5(15.6%) | 16(16.0%) | 22(14.1%) | 16(23.5%) | 6(12.8%) |
| Attended basic formal education | 19(23.8%) | 41(28.6%) | 13(25.0%) | 11(34.4%) | 27(27.0%) | 48(30.8%) | 17(25.0%) | 15(31.9%) |
| Attended secondary school | 27(33.8%) | 42(29.4%) | 12(23.1%) | 7(21.9%) | 31(31.0%) | 42(26.9%) | 15(22.1%) | 12(25.5%) |
| Attended higher education | 21(26.2%) | 38(26.6%) | 12(23.1%) | 9(28.1%) | 26(26.0%) | 44(28.2%) | 20(29.4%) | 14(29.8%) |
| Marital Status: | | | | | | | | |
| Single | 51(63.8%) | 99(69.2%) | 30(57.7%) | 26(81.2%) | 61(61.0%) | 107(68.6%) | 45(66.2%) | 33(70.2%) |
| Ever married | 29(36.2%) | 44(30.8%) | 22(42.3%) | 6(18.8%) | 39(39.0%) | 49(31.4%) | 23(33.8%) | 14(29.8%) |
| Participant Occupation: | | | | | | | | |
| Farmer | 4(5.0%) | 18(12.6%) | 13(25.0%) | 2(6.3%) | 7(7.0%) | 20(12.8%) | 14(20.6%) | 5(10.6%) |
| Employed | 41(51.3%) | 58(40.6%) | 19(36.5%) | 10(31.3%) | 46(46.0%) | 65(41.7%) | 23(33.8%) | 18(38.3%) |
| Student | 14(17.5%) | 32(22.4%) | 7(13.5%) | 10(31.3%) | 19(19.0%) | 34(21.8%) | 17(25.0%) | 12(25.5%) |
| Unemployed | 21(26.2%) | 35(24.5%) | 13(25.0%) | 10(31.3%) | 28(28.0%) | 37(23.7%) | 14(20.6%) | 12(25.6%) |
| Place of residence: | | | | | | | | |
| Rural | 30(37.5%) | 64(44.8%) | 17(32.7%) | 14(43.8%) | 38(38.0%) | 71(45.5%) | 24(35.3%) | 23(48.9%) |
| Urban | 50(62.5%) | 79(55.2%) | 35(67.3%) | 18(56.2%) | 62(62.0%) | 85(54.5%) | 44(64.7%) | 24(51.1%) |
| Ethnicity: | | | | | | | | |
| Oromo | 35(43.7%) | 106(74.1%) | 31(59.6%) | 20(62.5%) | 48(48.0%) | 116(74.4%) | 41(60.4%) | 30(63.8%) |
| Amhara | 20(25.0%) | 8(5.6%) | 7(13.5%) | 5(15.6%) | 23(23.0%) | 9(5.8%) | 9(13.2%) | 7(14.9%) |
| Sidama | 14(17.5%) | 16(11.2%) | 6(11.5%) | 2(6.3%) | 17(17.0%) | 17(10.9%) | 9(13.2%) | 4(8.5%) |
| Wolaita | 5(6.3%) | 8(5.6%) | 6(11.5%) | 4(12.5%) | 6(6.0%) | 8(5.1%) | 6(8.8%) | 4(8.5%) |
| Others | 6(7.5%) | 5(3.5%) | 2(3.9%) | 1(3.1%) | 6(6.0%) | 6(3.8%) | 3(4.4%) | 2(4.3%) |
| Religion: | | | | | | | | |
| Orthodox | 47(58.8%) | 21(14.7%) | 13(25.0%) | 10(31.3%) | 57(57.0%) | 25(16.0%) | 15(22.1%) | 13(27.7%) |
| Muslim | 8(10.0%) | 79(55.2%) | 16(30.8%) | 12(37.5%) | 13(13.0%) | 85(54.5%) | 26(38.2%) | 22(46.8%) |
| Protestant | 23(28.7%) | 36(25.2%) | 20(38.5%) | 9(28.1%) | 28(28.0%) | 39(25.0%) | 23(33.8%) | 11(23.4%) |
| Catholic | 2(2.5%) | 7(4.9%) | 3(5.7%) | 1(3.1%) | 2(2.0%) | 7(4.5%) | 4(5.9%) | 1(2.1%) |
| Spouse Education: | | | | | | | | |
| Not attended formal education | 6(20.7%) | 9(20.4%) | 9(40.9%) | 1(16.7%) | 8(20.5%) | 10(20.4%) | 9(39.1%) | 5(35.8%) |
| Attended basic formal education | 11(37.9%) | 15(34.1%) | 7(31.8%) | 3(50.0%) | 16(41.0%) | 17(34.7%) | 7(30.4%) | 3(21.4%) |
| Attended secondary school | 5(17.3%) | 12(27.3%) | 4(18.2%) | 1(16.7%) | 6(15.4%) | 13(26.5%) | 4(17.4%) | 3(21.4%) |
| Attended higher education | 7(24.1%) | 8(18.2%) | 2(9.1%) | 1(16.7%) | 9(23.1%) | 9(18.4%) | 3(13.1%) | 3(21.4%) |

*(Continued)*

**Table 3.** (Continued)

| Variable | Current use (YES) | | | | Ever use (YES) | | | |
|---|---|---|---|---|---|---|---|---|
| | Alcohol | Khat | Tobacco | Cannabis | Alcohol | Khat | Tobacco | Cannabis |
| Spouse Occupation: | | | | | | | | |
| Farmer | 1(3.4%) | 6(13.6%) | 3(13.6%) | 1(16.7%) | 3(7.7%) | 7(14.3%) | 3(13.0%) | 2(14.3%) |
| Employed | 18(62.1%) | 24(54.6%) | 8(36.4%) | 3(50.0%) | 23(59.0%) | 26(53.1%) | 9(39.1%) | 6(42.9%) |
| Student | 0(0.0%) | 2(4.5%) | 11(50.0%) | 0(0.0%) | 0(0.0%) | 2(4.1%) | 0(0.0%) | 1(7.1%) |
| Unemployed | 10(34.5%) | 12(27.3%) | 0(0.0%) | 2(33.3%) | 13(33.3%) | 14(28.6%) | 11(47.9%) | 5(35.7%) |
| Children Status: | | | | | | | | |
| No | 57(71.2%) | 105(73.4%) | 33(63.5%) | 27(84.4%) | 69(69.0%) | 114(73.1%) | 49(72.1%) | 36(76.6%) |
| Yes | 23(28.8%) | 38(26.6%) | 19(36.5%) | 5(15.6%) | 31(31.0%) | 42(26.9%) | 19(27.9%) | 11(23.4%) |
| Number of children: | | | | | | | | |
| <four children | 25(86.2%) | 39(88.6%) | 16(72.7%) | 6(100.0%) | 33(84.6%) | 44(89.8%) | 17(73.9%) | 14(100.0%) |
| ≥four children | 4(13.8%) | 5(11.4%) | 6(27.3%) | 0(0.0%) | 6(15.4%) | 5 (10.2%) | 6(26.1%) | 0(0.0%) |
| Household size: | | | | | | | | |
| Small family | 19(27.7%) | 42(29.4%) | 14(26.9%) | 8(25.0%) | 23(23.0%) | 45(28.9%) | 18(26.5%) | 11(23.4%) |
| Medium family | 43(53.8%) | 56(39.2%) | 24(46.2%) | 16(50.0%) | 53(53.0%) | 64(41.0%) | 30(44.1%) | 23(48.9%) |
| Large family | 18(22.5%) | 45(31.6%) | 14(26.9%) | 8(25.0%) | 24(24.0%) | 47(30.1%) | 20(29.4%) | 13(27.7%) |
| Family income per year: | | | | | | | | |
| Low income | 35(43.8%) | 56(39.2%) | 25(48.1%) | 16(50.0%) | 43(43.0%) | 60(38.5%) | 34(50.0%) | 19(40.4%) |
| Medium income | 28(35.0%) | 60(42.0%) | 19(36.5%) | 11(34.4%) | 38(38.0%) | 66(42.3%) | 25(36.8%) | 19(40.4%) |
| High income | 17(21.2%) | 27(18.9%) | 8(15.4%) | 5(15.6%) | 19(19.0%) | 30(19.2%) | 9(13.2%) | 9(19.2%) |
| Source of family income: | | | | | | | | |
| Agriculture | 23(28.8%) | 67(46.8%) | 19(36.5%) | 12(37.5%) | 32(32.0%) | 71(45.5%) | 30(44.1%) | 19(40.4%) |
| Trade | 18(22.5%) | 21(14.7%) | 11(21.2%) | 9(28.1%) | 21(21.0%) | 24(15.4%) | 12(17.6%) | 10(21.3%) |
| Private business | 11(13.8%) | 24(16.8%) | 6(11.5%) | 2(6.3%) | 16(16.0%) | 25(16.0%) | 8(11.8%) | 5(10.6%) |
| Salary | 28(35.0%) | 31(21.7%) | 16(30.8%) | 9(28.1%) | 31(31.0%) | 36(23.1%) | 18(26.5%) | 13(27.7%) |
| Level of social support: | | | | | | | | |
| Low social support | 33(41.25%) | 54(37.8%) | 31(59.6%) | 18(56.25%) | 40(40.0%) | 56(35.9%) | 40(58.8%) | 24(51.1%) |
| Moderate social support | 46(57.50%) | 84(58.7%) | 20(38.5%) | 14(43.75%) | 57(57.0%) | 95(60.9%) | 27(39.7%) | 23(48.9%) |
| High social support | 1(1.25%) | 5(3.5%) | 1(1.9%) | 0 | 3(3.0%) | 5(3.2%) | 1(1.5%) | 0 |
| Any mental health conditions: | | | | | | | | |
| No | 32(40.0%) | 79(55.2%) | 20(38.5%) | 12(37.5%) | 42(42.0%) | 85(54.5%) | 27(39.7%) | 17(36.2%) |
| Yes | 48(60.0%) | 64(44.8%) | 32(61.5%) | 20(62.5%) | 58(58.0%) | 71(45.5%) | 41(60.3%) | 30(63.8%) |
| Family history of substance use: | | | | | | | | |
| No | 33(41.25%) | 63(44.1%) | 16(30.8%) | 6(18.75%) | 40(40.0%) | 68(43.6%) | 22(32.3%) | 13(27.7%) |
| Yes | 47(58.75%) | 80(55.9%) | 36(69.2%) | 26(81.25%) | 60(60.0%) | 88(56.4%) | 46(67.7%) | 34(72.3%) |
| Family history of mental illness: | | | | | | | | |
| No | 63(78.7%) | 123(86.0% | 39(75.0%) | 24(75.0%) | 77(77.0%) | 132(84.6%) | 52(76.5%) | 37(78.7%) |
| Yes | 17(21.3%) | 20(14.0%) | 13(25.0%) | 8(25.0%) | 23(23.0%) | 24(15.4%) | 16(23.5%) | 10(21.3%) |

health conditions (depression, anxiety, post-traumatic disorder, suicidal behavior), had significantly higher odds of substance use (AOR = 1.99, 95%CI: 1.05, 3.80, p = 0.036).

From the final adjusted multivariable model analysis age, religion, occupation, education, family history of mental illness, and source of family income, did not demonstrate an independent association with the outcome variable of "any substance use."

## Discussion

Overall, 48.1% and 72.5% of the participants reported lifetime and current substance use, respectively. Factors such as being male, having single marital status, experiencing mental

**Table 4. Multivariable logistic regression model output indicating predictors of substance use among young people in West Arsi, Ethiopia, 2023, (n = 424).**

| Variable | Yes, for substance use (in %) N = 204 | COR 95%CI | P-Value | AOR, 95%CI | P-Value |
|---|---|---|---|---|---|
| Sex: | | | | | |
| Female | 20.1 | Ref | Ref | Ref | Ref |
| Male | 79.9 | 4.12(2.67, 6.35) | 0.000* | 6.37(3.35,12.11) | 0.000* |
| Age in years: | | | | | |
| 14–19 | 19.1 | Ref | Ref | Ref | Ref |
| 20–24 | 32.4 | 1.87(1.10, 3.18) | 0.021 | 1.34(0.63, 2.83) | 0.443 |
| 25–29 | 48.5 | 1.66(1.02, 2.70) | 0.042 | 1.48(0.62, 3.58) | 0.378 |
| Participant Education | | | | | |
| Not attended formal education | 17.2 | 1.00(0.51, 1.88) | 0.965 | 1.06(0.38, 2.93) | 0.906 |
| Attended basic formal education | 28.9 | 0.49(0.29, 0.83) | 0.007* | 0.48(0.21, 1.13) | 0.093 |
| Attended secondary school | 26.5 | 0.61(0.35, 1.04) | 0.072 | 0.62(0.27, 1.42) | 0.256 |
| Attended higher education | 27.4 | Ref | Ref | Ref | Ref |
| Marital Status: | | | | | |
| Single | 64.2 | 1.64(1.11, 2.42) | 0.013* | 2.12(1.06, 4.22) | 0.033* |
| Ever married | 35.8 | Ref | Ref | Ref | Ref |
| Participant Occupation: | | | | | |
| Farmer | 13.2 | Ref | Ref | Ref | Ref |
| Employed | 41.7 | 1.49(0.82, 2.69) | 0.187 | 1.41(0.54, 3.67) | 0.483 |
| Student | 20.6 | 0.88(0.46, 1.66) | 0.689 | 0.75(0.26, 2.13) | 0.587 |
| Unemployed | 24.5 | 1.32(0.70, 2.50) | 0.390 | 1.24(0.46, 3.36) | 0.669 |
| Place of residence: | | | | | |
| Rural | 43.1 | Ref | Ref | Ref | Ref |
| Urban | 56.9 | 1.20(0.82, 1.77) | 0.343 | 0.95(0.55, 1.64) | 0.853 |
| Religion: | | | | | |
| Orthodox | 28.9 | 3.36(1.98, 5.69) | 0.000* | 0.62(0.10, 3.93) | 0.617 |
| Islam | 43.6 | Ref | Ref | 0.22(0.04, 1.27) | 0.092 |
| Protestant | 23.5 | 1.32(0.83, 2.11) | 0.242 | 0.24(0.04, 1.45) | 0.120 |
| Catholic | 3.9 | 4.03(1.04,15.61) | 0.044* | Ref | Ref |
| Household size: | | | | | |
| Small family | 27.0 | Ref | Ref | Ref | Ref |
| Medium family | 45.6 | 0.63(0.39, 1.03) | 0.063 | 0.75(0.37, 1.50) | 0.414 |
| Large family | 27.4 | 0.71(0.41, 1.21) | 0.205 | 0.87(0.37, 2.02) | 0.744 |
| Source of family income: | | | | | |
| Agriculture | 47.7 | Ref | Ref | Ref | Ref |
| Trade | 15.7 | 1.50(0.83, 2.72) | 0.183 | 1.78(0.72, 4.39) | 0.209 |
| Private business | 15.7 | 0.73(0.43, 1.24) | 0.250 | 0.76(0.34, 1.70) | 0.511 |
| Salary | 25.5 | 1.48(0.90, 2.44) | 0.120 | 1.26(0.55, 2.89) | 0.580 |
| Level of perceived social support: | | | | | |
| Low | 34.3 | 16.67(6.30, 44.06) | 0.000* | 6.55(2.01, 21.30) | 0.002* |
| Moderate | 62.3 | 3.16(1.35, 7.43) | 0.008* | 2.40(0.88, 6.54) | 0.088 |
| High | 3.4 | Ref | Ref | Ref | Ref |
| Any mental health conditions: | | | | | |
| No | 52.9 | Ref | Ref | Ref | Ref |
| Yes | 47.1 | 2.54(1.81, 4.12) | 0.000* | 1.99(1.05, 3.80) | 0.036* |
| Family history of substance use: | | | | | |
| No | 44.6 | Ref | Ref | Ref | Ref |
| Yes | 55.4 | 13.14(1.69, 3.82) | 0.000* | 11.66(6.03, 22.54) | 0.000* |

*(Continued)*

**Table 4.** (Continued)

| Variable | Yes, for substance use (in %) N = 204 | COR 95%CI | P-Value | AOR, 95%CI | P-Value |
|---|---|---|---|---|---|
| Family history of mental illness: | | | | | |
| No | 83.3 | Ref | Ref | Ref | Ref |
| Yes | 16.7 | 2.55(1.36, 4.78) | 0.003[*] | 0.84(0.30, 2.37) | 0.750 |

[*]=P-value less than 0.05Ref= Referent category.

health conditions, having a family history of substance use, and lower perceived social support were associated with substance use among young people. The findings provide a broad analysis of the context relevant to substance use amongst young people in the West Arsi general community, Ethiopia.

The prevalence of substance use identified in this study was similar to that found in previous Ethiopian studies among young people, 47.0% [52], and a review of studies examining substance use among adolescents in East Africa, 49.0% [53]. However, the prevalence of substance use in this study was lower compared to various previous Ethiopian and international studies. For instance, higher prevalences were reported among high school adolescents in Woreta town (65.0%) [54], young people in the Northern Shewa zone, 66.0% [55], students at Addis Ababa University (74.0%) [56], college students in Kenya, 70% [57], and Ghanaian young adults in Accra, 71% [58]. Conversely, the prevalence in this study exceeds that reported in previous studies among young people in Ethiopia, (32.0%) [26], students in Ethiopia, (38.0%) [19], street children in Jimma town, (39.0%) [59], Nigerian high school students, (32.9%) [60], and University Students in Sudan, (31.0%), [61]. Observed differences in substance use prevalence between this and previous studies could stem from various factors, including sociocultural influences, environmental conditions, and methodological disparities (such as sample size, study duration, and the specific target population).

Along with the Ethiopian sociocultural practices that encourage communal gatherings, environmental factors conducive to psychoactive substance production can likely increase the risk of substance use [7,62]. In accordance with this, some parts of the current study setting, particularly Negelle Arsi and Wondo woredas, are known for producing substances such as *Arekie* (a potent local alcohol drink) [63] and *Wondo Balache khat* (a popular variety of khat among many Ethiopians) [64]. This might increase the tendency of young people to engage in the cultivation and production of these substances thereby increasing the risk of substance use, given over 85% of the population in this area is dependent on agricultural activities [65].

The geographic location of the study setting and unique cultural practices might also be attributed to the observed prevalence of substance use in this study. For example, Shashemene town (the zonal capital of the West Arsi zone) is located at a crossroads for the major roads that connect various parts of the country as well as with some neighboring countries such as Kenya [66]. As the transient population, particularly long-distance drivers often time engage in substance use, young people in the town might be more likely to be exposed to substance use and drug trafficking [67]. Furthermore, Shashemene is known to host a large community of *Rastafarians*, a religious group of Jamaican origin, which considers the town a *Land of Promise* [68,69]. The communal gatherings of this community are marked by activities such as smoking cannabis (ganja), which additionally exposes young people to substance use [70].

Moreover, the specific characteristics of the target population in this study may account for the higher prevalence of substance use observed. Substance use and its consequences are particularly pronounced in young people, as this age marks a critical developmental stage characterized by significant physical and psychological changes [71–73]. During this

transitional phase from childhood to adulthood, individuals often engage in risk-taking and sensation-seeking behaviors. This pursuit of novelty and excitement may lead to experimentation with substances. As postulated by the affect heuristic theory, decision-making during this period is often driven by emotions rather than rational analysis [74].

Consistent with previous Ethiopian [24,26,54] and international research [75–78], this study shows significant gender differences in substance use, with males showing high risk as compared to females. Ethiopian studies highlighted the gendered disparities in access to, use of, and control over resources, economic assets, physical mobility, decision-making power, and community expectations [79, 80]. They explained a different social definition and understanding of 'being young' for men and women, that young women in Ethiopia are part of the 'inner circle' of their parent's lives. Compared to young men, they have less or no leisure time and take on adult responsibilities earlier in life, such as taking on more household responsibilities and looking after younger siblings. In contrast, young men in Ethiopia have a 'public life' – they leave the house and spend their leisure time with friends engaging in activities that could expose them to different substances (for instance chewing khat or enjoying alcoholic drinks). These gendered socialisation norms in Ethiopia, encourage boys to be assertive and risk-taking, traits often associated with substance use experimentation [81]. This cultural practice of imposing more restrictions on females, particularly during their early age, compared to males, is common in the Arsi community, the dominant tribe in the West Arsi zone [82–84]. This tradition of associating substance use with masculinity, while stigmatising it among females as a deviation from traditional gender roles may have contributed to the observed high prevalence among males in our study. Additionally, young men may perceive substance use as a path to enhance their social status or peer conformity [12,52].

The current study found a significant association between substance use and marital status, with being single found to increase the odds of substance use. This finding aligns with previous studies conducted in Ethiopia and other countries [30,85–87]. In Ethiopia, marriage is a significant cultural institution, associated with heightened levels of family responsibility and personal security [88,89]. Having someone to share responsibilities, provide emotional support, and offer guidance can reduce feelings of isolation and stress, which are factors that may contribute to substance use. Married individuals may feel a greater sense of responsibility toward their spouse and family, which could protect them from engaging in risky behaviors like substance use [90]. Conversely, single individuals may have more social interactions outside of familial circles, potentially exposing them to environments where substance use is more common [52]. However, it is important to note that social desirability bias may play a role here with participants who are married potentially underreporting substance use due to societal pressure and fear of judgement.

Our study found experiencing mental health conditions like depression, anxiety, PTSD, and suicidal behaviors increased the odds of substance use among young people, consistent with previous studies [76,78]. This association likely arises from changes during adolescence and underlying stressors such as poverty, unemployment, violence, or social marginalization. Individuals with mental health conditions may turn to substances as self-medication or coping behaviour providing temporary relief from psychological pain or discomfort [71,91]. Limited access to mental health resources and the stigma surrounding mental health conditions may further drive young people to substance use as a temporary relief [92]. However, substance use can in some cases further disrupt neurobiological pathways, exacerbating existing mental health problems or triggering the emergence of new ones [93].

It was also found that a history of family substance use is a predictor of substance use prevalence among young people in the West Arsi zone, consistent with previous research [54,87,94]. Consistent with social learning theory, children learn behaviours by observing and

imitating family members [95,96]. When family members engage in substance use, it normalizes the behaviour, making young people more likely to view substance use as acceptable or desirable, thus prompting experimentation [78,97]. Furthermore, family environments with substance may lack supervision, discipline, and support, potentially leading to stress and maladaptive coping like substance use among young people, frequently in the context of parental conflict or neglect [78,98].

The current study also found higher substance use prevalence among participants with lower perceived social support, compared to those with moderate to higher support, highlighting the potential influence of social support on substance use patterns. This aligns with previous findings from Ethiopia [52,99] and elsewhere [100–102], indicating that individuals who receive higher levels of social support were at lower odds of reporting substance use. The extensive support networks within Ethiopian communities often play a vital role in an individual's life [103,104], and enhancing the protective role of these social networks in addressing substance use could be beneficial.

Overall, this substance use research highlights the importance of targeted interventions addressing substance use among young people focusing on promoting their mental well-being, enhancing their coping skills to different life challenges, and the importance of strengthening the family and social support dynamics.

## Strength and limitations

A major strength of this study is the comprehensive exploration of factors that may shape substance use behaviors such as gender, marital status, mental health conditions, family history of substance use, and perceived social support. Further, Our research methodology ensured broad participation regardless of literacy levels, employing face-to-face interviews with a standard questionnaire. We thoroughly selected participants to ensure diversity and achieved a high response rate (99.3%), indicative of strong engagement. Importantly, our approach minimized self-selection bias, enhancing the reliability and inclusivity of our findings.

A major limitation of this study pertains to its cross-sectional nature, limiting the establishment of causal relationships between variables. Additionally, as with any survey-based research, there is a potential for self-reporting bias, where participants may underreport or overreport substance use behaviors due to social desirability or other factors that could impact the accuracy and interpretation of results. Furthermore, it is important to note the finding of this study is specific to the West Arsi zone and might not entirely reflect substance use patterns in other regions of Ethiopia or different socio-cultural contexts.

## Conclusion and recommendations

In conclusion, this study sheds light on the prevalence and associated factors of substance use among young people in the West Arsi zone, Ethiopia. The study provides valuable insights into the complexities of substance use behaviors among young people in the region.

Approximately half of the study participants reported at least one substance use in their lifetime from alcohol, khat, tobacco, and cannabis, with variations across different factors such as gender, marital status, mental health conditions, family history of substance use, and perceived social support. These findings underscore the need for targeted interventions and public health policies that address the specific needs of young individuals and the diversity of substances used in the West Arsi zone and similar regions in Ethiopia.

Targeted community-based intervention strategies are essential to address substance use among young people in Ethiopia, focusing on their mental well-being, fostering their coping

skill development, and strengthening family and social support systems as positive influences on their lives. Additionally, comprehensive research is needed to deepen understanding of substance use among young people in Ethiopia, exploring diverse risk factors, causality, associated harms, and intervention effectiveness, facilitating more informed and effective prevention and intervention efforts.

## Supporting information

**S1 Table:** Proportional distribution of sample size among the chosen kebeles, West Arsi zone, Ethiopia, 2023 (N = 427).
(DOCX)

## Acknowledgment

The authors would like to express their heartfelt gratitude to all study participants for their invaluable contributions to this research by sharing their information and experiences. Appreciation is also extended to everyone involved in the research process, including the committed data collectors, zonal and woreda administration officials and staff, and the research office at Madda Walabu University, Shashemene campus, for their vital support.

## Author contributions

**Conceptualization:** Jemal Ebrahim Shifa, Jon Adams, Daniel Demant.

**Formal analysis:** Jemal Ebrahim Shifa.

**Funding acquisition:** Jemal Ebrahim Shifa, Jon Adams, Daniel Demant.

**Investigation:** Jemal Ebrahim Shifa.

**Methodology:** Jemal Ebrahim Shifa, Jon Adams, Daniel Demant.

**Project administration:** Jemal Ebrahim Shifa, Jon Adams, Daniel Demant.

**Resources:** Jemal Ebrahim Shifa.

**Software:** Jemal Ebrahim Shifa.

**Supervision:** Jemal Ebrahim Shifa, Jon Adams, Daniel Demant.

**Validation:** Jemal Ebrahim Shifa, Jon Adams, Daniel Demant.

**Visualization:** Jemal Ebrahim Shifa, Jon Adams, Daniel Demant.

**Writing – original draft:** Jemal Ebrahim Shifa.

**Writing – review & editing:** Jemal Ebrahim Shifa, Jon Adams, Daniel Demant.

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
