## [Decision Letter · Decision Letter 0]

4 Dec 2024

Dear Dr. Shifa,

Thank you for submitting your manuscript to PLOS ONE. After careful consideration, we feel that it has merit but does not fully meet PLOS ONE’s publication criteria as it currently stands. Therefore, we invite you to submit a revised version of the manuscript that addresses the points raised during the review process.

We look forward to receiving your revised manuscript.

Kind regards,

Zakir Abdu (Assistant Professor)

Academic Editor

PLOS ONE

Journal Requirements:

2. In the online submission form, you indicated that the data supporting the findings of this study are included in the manuscript. Further information on the data and materials used in this study can be obtained from the corresponding author upon reasonable request.

Additional Editor Comments:

1. How you get informed consent for the participant’s age below 18?

2. The response rate was high which is unusual for survey

3. What is do you think on multiple substance use?

4. You were tried to include the participants based on the duration of they live in the area. 6 months and above. Justify scientifically

5. It is well if you elaborate the eligibility criteria

6. Both single and double population formula was used for sample size calculation. Why double population?

7. A convenient and purposive sampling method was applied to select the study setting,…. Not clear

8. You were selected four woredas from the total. How and based on what? And four kebeles from woreda. Not clear

9. You were gave ID or number for house and it is good. But, from where you started and ended? I think the total house of per kebeles also important

10. Sometimes there might be more than one participants were live in single home and how did you were approached?

11. Analysis of ethnicity and religion in relation to prevalence of substance use and associated factors is sensitive issue.

12. Think on the results. Or elaborate and make clear. The life time prevalence is less than the current one. It looks as the current users also lifetime users.

13. Why if you run associated factors for all use category (lifetime, current, and problematic use)

14. For the discussion part, well if you only put the justification of findings rather than comparing with previous findings

Reviewers' comments:

Reviewer's Responses to Questions

**Comments to the Author**

1. Is the manuscript technically sound, and do the data support the conclusions?

Reviewer #1: Yes

Reviewer #2: Yes

2. Has the statistical analysis been performed appropriately and rigorously?

Reviewer #1: Yes

Reviewer #2: Yes

3. Have the authors made all data underlying the findings in their manuscript fully available?

Reviewer #1: Yes

Reviewer #2: Yes

4. Is the manuscript presented in an intelligible fashion and written in standard English?

Reviewer #1: Yes

Reviewer #2: Yes

Reviewer #1: Title: Substance use among young people in the West Arsi Zone, Ethiopia: A cross-sectional study

Manuscript Number: PONE-D-24-45776

Article Type: Research Article

Comments for the authors

This research establishes a prevalence and factors associated with substance use among young population aged 14-29years in the West Arsi zone. The study presented an overall lifetime prevalence of any substance use among the study participants was 48.1%, with 76.5% reporting khat chewing, 49.0% alcohol drinking, 33.3% use of different forms of tobacco, and 23.0% cannabis use. A study also highlighted Being male, having a single marital status, a family history of substance use, low perceived social support, and the presence of mental health conditions as factors associated with an increased likelihood of substance use.

This study is an important contribution to the field of mental health. it is community based studies targeting youth populations and their risk for substance use. I would like to congratulate authors on the paper for their great work in writing this paper and can be published in its current form addressing few concerns stated below.

1. A statement in the abstract section …. ‘‘33.3% use of different forms of tobacco’’ needs further explanation. This is big figure it has to specifically for types of tobacco consumed in the local community.

2. What does mean ‘‘mental health conditions’’ in the current study. Since the presence of the underlying mental health conditions with was even found to have significant association with outcome of the interest. Operationalization of this terms has to be added for enhancing readability of the paper.

3. Table must be cell-based and has to be in line as per the guidelines of the journal.

Reviewer #2: How can current substance use be higher than lifetime use? You should reconsider your findings. What is the value of using the AUDIT if you’re not conducting an analysis? Why not analyze current substance use instead? Current or recent use is more relevant, whereas lifetime use may not provide the same insights. Please clarify the implications of this approach.

**Do you want your identity to be public for this peer review?** For information about this choice, including consent withdrawal, please see our Privacy Policy

Reviewer #1: No

Reviewer #2: **Yes: ** Yacob Abraham Borie.

---

## [Author Response · Author response to Decision Letter 1]

19 Dec 2024

Reviewer 1

Comment#1: A statement in the abstract section …. ‘‘33.3% use of different forms of tobacco’’ needs further explanation. This is big figure it has to specifically for types of tobacco consumed in the local community.

Response#1: Thank you for your suggestion. Including the specific forms of tobacco used (both for current and lifetime use) in the abstract may lengthen the abstract and potentially condense other critical information. Rather, we have ensured that the detailed breakdown of both current and lifetime forms of tobacco use is comprehensively presented in the results section under the substance use subsection on page 10 of the manuscript. This approach helps balance conciseness and detail while adhering to abstract length requirements. Thank you for your understanding and consideration.

FYI: Lifetime tobacco use was calculated among individuals who reported lifetime use of any substance in the list (68/204), whereas current use was derived from lifetime users (52/68).

Comment#2: What does mean ‘‘mental health conditions’’ in the current study. Since the presence of the underlying mental health conditions with was even found to have significant association with outcome of the interest. Operationalization of this terms has to be added for enhancing readability of the paper.

Response#2: Thank you again for raising this important issue: In this study, four mental health conditions—depression, anxiety, PTSD, and suicidal behavior—were independently screened using validated tools. The PHQ-9 was used for depression, GAD-7 for anxiety, TSQ-10 for PTSD, and items adapted from prior studies for assessing suicidality. Participants were categorized as positive for mental health conditions if they screened positive for at least one of these disorders, otherwise classified as negative. Detailed descriptions of these assessments are provided in the methods section, specifically under "Variables and Measures," on page 10 of the revised manuscript version.

Comment#3: Table must be cell-based and has to be in line as per the guidelines of the journal.

Response#3: Accepted and changed as per the suggestion

Reviewer 2

Comment#1: How can current substance use be higher than lifetime use? You should reconsider your findings. Response#1: Thank you for catching up on this difference.

In our study, lifetime prevalence refers to the proportion of individuals who have ever used a substance at any point in their life up to the time of the survey, which in our case is 48.1% (204/424). This figure represents those who have ever used any of the substances. On the other hand, current prevalence refers to the proportion of individuals who used a substance within the 30 days preceding the survey. This category is a subset of lifetime use, as it includes only those who are actively using substances within that timeframe.

The current prevalence among lifetime users is 72.5%, calculated as 148/204. This means that 72.5% of the lifetime users were actively using substances in the 30 days prior to the survey. It’s important to note that this 72.5% applies only to the 204 lifetime users, not the entire sample. This distinction is consistent across all the numbers representing current and lifetime substance use prevalence in our study.

Again, this part under the abstract and results section, was thoroughly revised for clarity. See the abstract page and pages 14 & 15 under the result section of the revised manuscript, please.

Comment#2: What is the value of using the AUDIT if you’re not conducting an analysis?

Response#2: Thank you for this important question. Part of the analysis derived from the AUDIT tool is presented in the results section, including risk categorization into low risk, probable hazardous use, and probable alcohol dependence, as detailed on page 14 of the revised manuscript. While this manuscript focuses on providing an overview, we plan to explore the AUDIT results in greater depth in a subsequent alcohol-specific paper to provide a comprehensive analysis.

Comment#3: Why not analyze current substance use instead? Current or recent use is more relevant, whereas lifetime use may not provide the same insights. Please clarify the implications of this approach.

Response#3: Thank you for this important question: We firmly acknowledge analyzing prevalence and associated factors across all use categories—lifetime, current, and problematic use—offers valuable insight into substance use patterns and their determinants. Each dimension uniquely contributes to understanding substance use behaviors and aligning appropriate intervention strategies.

Lifetime use captures the widest group, including those who may no longer use substances but whose behaviors were influenced by past factors. Current use focuses on active users, identifying immediate risks and needs. Problematic use zeroes in on individuals with dependency or harmful patterns, guiding more intensive interventions.

For example:

• Lifetime use analysis informs strategies to prevent initiation.

• Current use analysis targets interventions to reduce ongoing consumption.

• Problematic use highlights the need for treatment and rehabilitation.

While presenting all categories together is not mandatory, it is firmly believed that differentiating between them can be essential for tailored and layered intervention designs. In line with this, our study reported on descriptive findings for both current and problematic use, whereas for lifetime substance use, we included both descriptive and inferential analyses to provide a comprehensive understanding of its associated factors and prevalence. This approach balances the need for immediate insights with the potential for more tailored, evidence-based interventions.

For detailed information on this, please refer to the abstract and results section, which was thoroughly revised for clarity. See the abstract page and pages 14 & 15 under the result section of the revised manuscript, please.

Response for the editor/s

Comment#1: How do you get informed consent for the participant’s age below 18?

Response#1: Consent and Assent Process:

• Participants 18 and above: provided informed consent themselves.

• Participants below 18 years:

Parental/guardian consent: Obtained from a parent or legal guardian.

Assent: Obtained from the minor, ensuring they voluntarily agree to participate after an age-appropriate explanation of the study.

As part of the process,

We prepared forms that include all relevant information about the study in a way and language that is understandable to participants.

We ensure that minors understand the study's purpose, procedures, and their rights, including the ability to withdraw at any time without consequences.

We maintain confidentiality throughout the data collection phase.

We strictly adhere to local and international ethical guidelines.

Please, see pages 7 and 8 of the revised manuscript, under the method section.

Comment#2: The response rate was high which is unusual for a survey

Response#2: Thank you for raising this important question. The high response rate (99.3%) can be attributed to several factors. The data collection was conducted face-to-face, allowing researchers to establish direct rapport with participants, which is known to encourage higher participation. Local leaders were actively involved in the process, fostering trust and credibility within the community. Moreover, we ensured that data collectors were familiar with the study setting and proficient in the local languages, which minimized potential communication barriers and increased participant engagement.

Additionally, we adhered to ethical protocols by obtaining parental consent for participants under 18 years of age, as well as participant assent. This inclusive and respectful approach likely enhanced willingness to participate. We can confidently confirm that it is attributed to a well-executed engagement strategy rather than any form of selection bias or data errors.

Comment#3: What do you think about multiple substance use?

Response#3: Thank you for this insightful question: Our study defines substance use prevalence—whether lifetime or current—based on a "yes" response to at least one of four substances: alcohol, khat, tobacco, or cannabis. While we acknowledge the significance of multiple substance use, our primary aim was to establish baseline prevalence and identify associated factors. We recommend future studies to explore patterns of poly-substance use for more comprehensive insights and intervention strategies.

Comment#4: You tried to include the participants based on the duration of they lives in the area. 6 months and above. Justify scientifically

Response#4: Thank you for this question. Including participants based on a minimum residency duration of six months in the area is a well-established practice in community-based research. In the Ethiopian context, six months is widely regarded as the minimum period required for recognizing residency in a neighborhood. This criterion ensures that participants have had sufficient exposure to the local environment, social norms, and available resources, which may influence behaviors, health outcomes, and experiences, including substance use and mental health conditions.

Scientifically, this approach helps ensure representativeness, as individuals with a longer duration of residence are more likely to reflect the characteristics and experiences of the community, enhancing the generalizability of the findings. Moreover, it reduces potential bias by excluding short-term residents who may not have fully adapted to the local context or been influenced by its unique factors. This criterion also aligns with standard methodologies in similar public health and behavioral studies.

This is detailed in the revised manuscript under the method section on page 5 under study population and recruitment.

Comment#5: It is well if you elaborate on the eligibility criteria

Response#5: The study population for this research consisted of young people aged 14-29 years living in the West Arsi zone, Oromia region, Ethiopia. This age group was selected for its demographic dominancy within the country and its representation of a critical developmental stage where substance use, mental health conditions, and quality of life are priority concerns.

Inclusion criteria: young people aged 14-29 years, current residents of the study area who had lived there for at least six months, could provide informed consent and/or assent, and were willing to participate.

Exclusion criteria: Young people who were critically ill and unable to provide information and individuals absent during the data collection period were excluded.

This is well detailed in the revised manuscript under the method section on page 5 under study population and recruitment.

Comment#6: Both single and double population formula was used for sample size calculation. Why double the population?

Response#6: We used the double population formula to calculate the sample size for subgroup analyses, particularly for factors previously identified as associated with substance use, such as sex, family history of substance use, and place of residence (rural/urban). This approach ensures that the study is adequately powered to detect significant differences within these subgroups.

Then, after calculating the required sample size using both the single and double population formulas, we selected the larger to ensure a sufficient sample size for robust subgroup analyses and overall increased statistical power. We firmly believe that this evidence-based methodological approach enhances the reliability and validity of our study findings while supporting meaningful comparisons across key variables.

For clarity, we have revised this section which can be referred to on page 5 of the revised manuscript.

Comment#7: A convenient and purposive sampling method was applied to select the study setting,…. Not clear Response#7: The West Arsi Zone was selected purposively due to its alignment with the study's objectives and the primary researcher's familiarity with the area. Factors such as the availability and accessibility of various substances, the demographic dominance of young people, and a high unemployment rate made it a relevant setting for the research.

Woredas (districts) within the zone were selected randomly, and the kebeles were selected conveniently, primarily based on their accessibility and the concentration of the young population. This combination of purposive and convenience sampling ensured that the study setting was both relevant to the research questions and logistically feasible for data collection.

This part is thoroughly revised for clarity as on page 6 of the revised manuscript.

Comment#8: You were selected four woredas from the total. How and based on what? And four kebeles from Woreda. Not clear

Response#8: We selected four woredas from the total 13 in the West Arsi Zone, representing approximately 30% of the woredas. This proportion is considered scientifically sound for ensuring representativeness while maintaining feasibility. A similar procedure was followed for selecting kebeles within each woreda, with criteria emphasizing population size, youth demographics, and accessibility to align with the study’s objectives.

Again, this method section under the subtitle study population and recruitment, was thoroughly revised for clarity. See pages 5 & 6 of the revised manuscript, please.

Comment#9: You were given an ID or number for the house and it is good. But, from where do you start and end? I think the total house per kebeles is also important

Response#9: Yes, we assigned a unique identification number to each household in the selected kebeles to establish a comprehensive sampling frame. The numbering process began at a central or predefined landmark in each kebele, such as the kebele administration office, mosque, church, school, or another prominent feature depending on the setting. From this starting point, households were numbered systematically, ensuring full coverage within the kebele boundary until the last household to be included in the study was reached.

The total number of households in each kebele was also recorded to ensure that the sampling frame was complete and accurate. This total served as the foundation for selecting households proportionally or randomly, ensuring representativeness and minimizing selection bias.

Again, this method section under the subtitle study population and recruitment, was thoroughly revised for clarity based on your feedback. See pages 6 & 7 of the revised manuscript, please.

Comment#10: Sometimes there might be more than one participant were lives in a single home how did you were approached?

Response#10: Data collectors selected and interviewed one eligible individual per household, using a lottery method in those households where multiple individuals who met the study inclusion criteria were present.

See pages 6 & 7 of the revised manuscript for detailed information in this regard.

Comment$11: Analysis of ethnicity and religion in relation to the prevalence of substance use and associated factors is a sensitive issue.

Response#11: Thank you for this critical question. Analyzing ethnicity and religion in relation to substance use and associated factors is indeed a sensitive issue, particularly in Ethiopia, where diverse cultural, traditional, and spiritual practices play significant roles in shaping individuals' behaviors and societal norms. Addressing this topic requires cultural humility and ethical responsibility to avoid reinforcing stereotypes or stigmatization.

Key considerations include:

1. Cultural Context: Ethiopia's rich diversity means substance use and mental health conditions are deeply influenced by cultural, traditional, and spiritual practices. For instance, khat chewing is often linked to social and religious gatherings, and alcohol use may be tied to cultural ceremonies. Exploring these practices helps understand the social acceptability and accessibility of substances within specific groups.

2. Sensitivity and Respect: Discussions ab

---

## [Decision Letter · Decision Letter 1]

2 Feb 2025

Substance use among young people in the West Arsi Zone, Ethiopia: A cross-sectional study.

PONE-D-24-45776R1

Dear Mr. Ebrahim,

We’re pleased to inform you that your manuscript has been judged scientifically suitable for publication and will be formally accepted for publication once it meets all outstanding technical requirements.

Kind regards,

Zakir Abdu

Academic Editor

PLOS ONE

Additional Editor Comments (optional):

On the title add the year of study

Reviewers' comments:

Reviewer's Responses to Questions

**Comments to the Author**

Reviewer #1: All comments have been addressed

Reviewer #2: All comments have been addressed

2. Is the manuscript technically sound, and do the data support the conclusions?

Reviewer #1: Yes

Reviewer #2: Yes

3. Has the statistical analysis been performed appropriately and rigorously?

Reviewer #1: Yes

Reviewer #2: Yes

4. Have the authors made all data underlying the findings in their manuscript fully available?

Reviewer #1: Yes

Reviewer #2: (No Response)

5. Is the manuscript presented in an intelligible fashion and written in standard English?

Reviewer #1: Yes

Reviewer #2: Yes

Reviewer #1: (No Response)

Reviewer #2: no additional comments but I would like to thank for their adequate response and explanation for my questions and comments

**Do you want your identity to be public for this peer review?** For information about this choice, including consent withdrawal, please see our Privacy Policy

Reviewer #1: No

Reviewer #2: **Yes: ** Yacob Abraham Borie

---

## [Editor Report · Acceptance letter]

PONE-D-24-45776R1

PLOS ONE

Dear Dr. Shifa,

I'm pleased to inform you that your manuscript has been deemed suitable for publication in PLOS ONE. Congratulations! Your manuscript is now being handed over to our production team.

Kind regards,

on behalf of

Zakir Abdu

Academic Editor

PLOS ONE